# Online Adaptation of Language Models with a Memory of Amortized Contexts

**Jihoon Tack**[1], **Jaehyung Kim**[2], **Eric Mitchell**[3], **Jinwoo Shin**[1],
**Yee Whye Teh**[4], **Jonathan Richard Schwarz**[5]
[1]KAIST  [2]Yonsei University  [3]Stanford University
[4]University of Oxford  [5]Harvard University & Thomson Reuters
jihoontack@kaist.ac.kr

## Abstract

Due to the rapid generation and dissemination of information, large language models (LLMs) quickly run out of date despite enormous development costs. To address the crucial need to keep models updated, online learning has emerged as a critical tool when utilizing LLMs for real-world applications. However, given the ever-expanding corpus of unseen documents and the large parameter space of modern LLMs, efficient adaptation is essential. To address these challenges, we propose Memory of Amortized Contexts (MAC), an efficient and effective online adaptation framework for LLMs with strong knowledge retention. We propose a feature extraction and memory-augmentation approach to compress and extract information from new documents into compact modulations stored in a memory bank. When answering questions, our model attends to and extracts relevant knowledge from this memory bank. To learn informative modulations in an efficient manner, we utilize amortization-based meta-learning, which substitutes an otherwise required optimization process with a single forward pass of the encoder. Subsequently, we learn to choose from and aggregate selected documents into a single modulation by conditioning on the question, allowing us to adapt a frozen language model during test time without requiring further gradient updates. Our experiment demonstrates the superiority of MAC in multiple aspects, including online adaptation performance, time, and memory efficiency. In addition, we show how MAC can be combined with and improve the performance of popular alternatives such as retrieval augmented generations (RAGs). Code is available at: https://github.com/jihoontack/MAC.

## 1  Introduction

Language models (LMs) [7, 79] have significantly accelerated progress in natural language processing (NLP) and thus become a core technology in various real-world applications, such as coding assistants [10], search engines [90], and personal AI assistants [16]. However, LMs are typically static artifacts, and as the world changes, the knowledge encoded in their parameters becomes outdated. This becomes especially problematic for large language models (LLMs), as multiple applications (e.g., Chatbots [34, 55]) require the model to be up-to-date, yet retraining LLMs with new documents from scratch requires high computational demands [31].

To tackle this issue, multiple studies suggested online and continual learning frameworks for LMs, i.e., adapting the LM on a stream of new documents. One line of work proposes to use retrieval-augmented models by saving the stream of documents and selecting the most relevant document based on the input [9, 33]. However, even large models often fail to update their learned knowledge when the retrieved document consists of counterfactual information [48, 44, 75] and it may not be suited for

38th Conference on Neural Information Processing Systems (NeurIPS 2024).

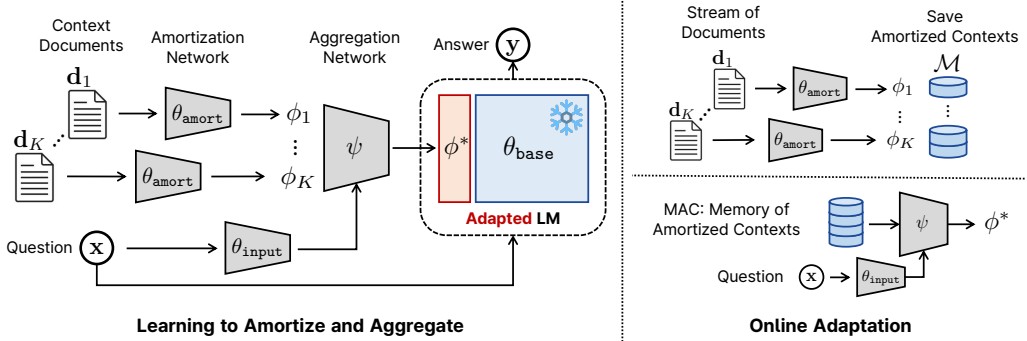

Figure 1: An overview of MAC: we amortize each context document into PEFT modulation $\phi$ and learn to aggregate modulations into a single target modulation $\phi^*$ based on the given question input $\mathbf{x}$ to adapt the frozen LM $\theta_{\texttt{base}}$. During online adaptation, we store the amortized contexts into a memory bank $\mathcal{M}$, then adapt the LM via aggregating the memory bank based on the given question.

edge computing as a large number of documents poses expensive computation for model inference [26]. Due to these limitations, another line of recent works suggests finetuning the model on a stream of documents to directly update the knowledge inside the LM (i.e., online finetuning [42, 32]). While effective, online finetuning schemes also face limitations such as a large computation for gradient calculation, the sensitivity of the online optimization hyper-parameter [26], and the aforementioned catastrophic forgetting problem [50, 39]. In this paper, we instead ask: *Can we tackle the limitations of retrieval augmented models and online finetuning by assimilating and retaining knowledge from incoming documents without the need for gradient-based learning at test time?*

To this end, we suggest bridging this gap through a complementary learning systems approach [41] by introducing an end-to-end differentiable auxiliary retrieval augmentation system that can be run alongside a (frozen) target LM. This system extracts knowledge from incoming documents, builds a memory bank, and learns to automatically select relevant information from this memory bank, which is subsequently passed as additional input to the target model. Once learned, this system can be effectively employed purely through forward passes.

**Contribution.** We propose Memory of Amortized Contexts (MAC), an efficient and effective online learning framework for LMs (see the overview in Figure 1). The core idea of MAC is to freeze LM parameters (thus reducing undesirable side effects common for online finetuning) and instead incorporate new information through additional learned input tokens (an established Parameter-Efficient Fine-Tuning technique [47]), utilizing amortization-based meta-learning [19, 65]. Specifically, instead of optimizing individual PEFT tokens (which necessitates labels and gradient computations), we instead learn to directly predict these tokens based on a query and memory bank alone, without the need for labels at test time, thus proposing amortized optimization [1, 49].

To ensure the scalability of MAC, we propose two memory-efficient techniques for training and inference: (1) We find that the process of training our complementary retrieval and aggregation operation for LLMs, necessitates a sufficiently large batch size, which introduces significant memory constraints. To address this issue, we backpropagate on only a random subset of documents, significantly saving memory while still providing an unbiased approximation of the full gradients [6]. (2) Large memory banks can further increase GPU memory usage when aggregating information relevant to a query during inference. To address this, we propose a divide-and-conquer approach, sub-grouping the large set of modulations into smaller, manageable groups and repeating this procedure with the predicted modulations until the final modulation parameters are determined.

We verify the efficacy of MAC through evaluations on multiple datasets and architectures. Overall, our experimental results demonstrate the strong results of MAC. For instance, when measured with the F1 score (%), MAC improves performance from $18.97 \rightarrow 21.79$ over prior work on StreamingQA [45], and $18.66 \rightarrow 21.14$ on SQuAD-Seq [26]. Furthermore, we demonstrate that MAC shows significant effectiveness in retaining learned knowledge when compared to other online finetuning baselines, justifying the memory-augmentation approach. In addition, MAC can be readily combined with retrieval augmented generation (RAG) and in effect, further increases the selection quality of retrieved documents, resulting in an improvement of $71.83 \rightarrow 74.89$ over BM25 alone [66] on ArchivalQA-Seq. Finally, we highlight the efficiency of MAC in multiple aspects, measuring adaptation time, training, and inference memory usage, again demonstrating strong improvements over baselines.

## 2 Related Work

**Amortization-based meta-learning.** Amortization-based meta-learning, which encodes the given context to directly predict the task-specific model, has gained much attention due to its computational efficiency as it only requires a single encoder forward pass when adapting the model [69, 51, 19, 18]. These approaches, especially when combined with modulation techniques, have achieved notable success in various applications, such as few-shot visual recognition [65, 6, 11] and 3D reconstructions [20, 35]. Recently, this idea has been extended to language domains where prior works facilitate hypernetworks to adapt LMs with given few-shot prompts [58, 28]. In this paper, we extend the use of amortization-based meta-learning to extract the knowledge of a given document into a compact yet informative modulation for online adaptation.

**Online learning.** Online learning, also referred to as continual or lifelong learning, is a task of adapting models to new data or task distributions [77]. Such ideas are becoming increasingly relevant in the era of deep learning generally and with the advent of extremely large models [78, 17, 71] specifically. In the language domain, there have been various attempts to tackle online learning [40, 92, 63] where recent studies focus more on online learning of LLMs, e.g., finetuning on a stream of documents [42], architectural constraints [32], and the use of replay buffers [14]. Among them, Hu et al. [26] found that online finetuning can be effective when an LM focuses on important tokens during the adaptation and proposed a gradient-based meta-learning approach to automatically learn a token importance weighting model. However, such gradient-based meta-learning schemes require a compute-expensive second-order gradient calculation [15, 64]. Moreover, online finetuning schemes can face multiple challenges, including (i) inevitable forgetting of the learned knowledge, (ii) gradient computation of LLMs during adaptation, and (iii) high sensitivity to the online optimization hyperparameter (e.g., learning rate [26]). MAC does not suffer from such issues as our amortization strategy is efficient without introducing any hyperparameters while effectively preserving knowledge.

**Retrieval augmentation for LMs.** Retrieval augmentation of LMs with relevant information from external knowledge sources has served as an effective way to improve the performance of LMs on various NLP tasks [21, 43, 30, 70, 80] by reducing hallucination and leveraging external knowledge which is not seen during pre-training. However, retrieval augmentation drastically increases computational cost [88] as documents often consist of thousands of words. In addition, its effectiveness is sensitive to the configuration of retrieved information [46], and even negatively affects the performance of LMs when the retrieved information is counterfactual [75]. MAC is more efficient than retrieval augmentation as it amortizes the external knowledge to modulate LMs rather than directly incorporating it. Furthermore, we believe MAC and retrieval augmentation has similarities as both methods store the knowledge and utilize them base on the user query, while the main difference is that MAC attend to multiple documents simultaneously using the aggregation network, allowing the LLM to capture shared information across documents. We thus believe that the joint usage benefits retrieval augmentation, as MAC can guide retrieval augmentation to capture missing information not retrieved by the retriever (see Section 4.1 for the supporting experiment).

**Memory augmented LMs.** Recently, memory augmentation has also shown great promise for LMs where it significantly improves the performance and efficiency in various directions [84, 56, 94, 54, 24], e.g., extending context length with memory retrieval [87, 83], personalization [2], and model editing [53]. Unlike these methods, which store the raw text or use the memory bank to train new LMs, MAC stores compact modulation parameters (in the shape of learned tokens) and adapts the frozen target LM, thereby utilizing large models without the heavy computation of training LMs.

## 3 MAC: Online Adaptation with a Memory of Amortized Contexts

In this section, we first briefly describe our problem setup (Section 3.1), then core components, namely amortization and aggregation framework (Section 3.2) and finally, efficient training and inference schemes for MAC (Section 3.3). Algorithm 1 and 2 in Appendix B provide detailed training and online adaptation processes for our framework.

### 3.1 Problem setup: Online adaptation

We consider the online adaptation scenario proposed in Hu et al. [26] where a static LM parameterized by $\theta_{\texttt{base}}$ is adapted to an online stream of documents $\mathcal{C}^{\texttt{test}} := (\mathbf{d}_1, \cdots, \mathbf{d}_{K^{\texttt{test}}})$. After incorporating

the final document, we then evaluate the adapted model's performance with a set of queries $\{\mathbf{x}_i\}$ and a corresponding labels $\{\mathbf{y}_i\}$, where the $i^{\text{th}}$ query and label are drawn from a conditional distribution of a document $\mathbf{d}_i$, i.e., $(\mathbf{x}_i, \mathbf{y}_i) \sim p(\mathbf{x}, \mathbf{y}|\mathbf{d}_i)$. Here, note that the query $\mathbf{x}_i$ is not accessible during online adaptation; hence, retaining the learned information from $\mathbf{d}_i$ is critical for achieving good results. While the query input and label pair $(\mathbf{x}, \mathbf{y})$ can be in any format or task, we mainly focus on question and answering (QA) tasks by following Hu et al. [26], i.e., $\mathbf{x}_i$ is a question and $\mathbf{y}_i$ is the corresponding answer based on the given information in $\mathbf{d}_i$, as it is straightforward to evaluate the LM's updated knowledge. Nevertheless, we also consider an additional non-QA setup in Section 4.3.

## 3.2 MAC: Memory of amortized contexts

The stated goal of MAC is (i) the efficient adaptation of a given LM to unseen information (ii) while retaining previously learned knowledge, both from its original training stage as well as updates from prior examples in a stream of novel data. To this end, we propose to utilize amortization-based meta-learning [18, 19] of a memory-augmented system. Amortization-based meta-learning with *modulations* [27, 65, 4] learns to predict a task-specific modulation (i.e., a compact representation of a task) through amortizing the given context set sampled from the task distribution. This enables efficient adaptation using the learned amortization network, as it only requires a single forward pass to adapt a model, foregoing the cost of gradient computation. It is worth noting that this is also beneficial as the LM does not have access to the input and label pair $(\mathbf{x}, \mathbf{y})$ during the online adaptation, where we can design the amortization to find the modulation only with the given document $\mathbf{d}$. Furthermore, meta-learned modulations have been found to preserve the task information well (e.g., showing great potential for generating or classifying distributions of tasks [72, 73]). They can hence be expected to effectively extract document information. Based on this insight, we suggest meta-learning the amortization network to directly predict a compact modulation for a new document.

**Learning to amortize contexts.** For a given context document $\mathbf{d}_k$ sampled from the training document set $\mathcal{C}^{\text{train}}$, we learn an amortization network parameterized by $\theta_{\text{amort}}$ to predict a modulation parameter (of the same shape as embedded tokens) $\phi_k$ as: $\phi_k := g_{\theta_{\text{amort}}}(\mathbf{d}_k)$. Here, we use a hypernetwork [22] for $\theta_{\text{amort}}$: we modify the T5 architecture [60] by having learnable tokens as the input of the decoder to have a consistent number of output tokens by following [58]. One can design the modulation with any type of PEFT scheme (e.g., LoRA [25] or FiLM [57]), among which we use P-Tuning v2 [47] (i.e., predictions of the key-value of each attention layer).

**Modulating LMs via aggregating amortized contexts.** Given a memory bank of compressed documents in the form of modulations $\{\phi_k\}_{k=1}^K$, we now learn to choose relevant information in the form of a modulation $\phi_i^*$ for a given input $\mathbf{x}_i$. While one design choice is to select/retrieve a single modulation, this has two drawbacks: (i) risk of selecting the wrong modulation and (ii) limited utilization of learned knowledge across different modulations. Moreover, it is worth noting that recent studies empirically show that linear interpolation (or advanced merging) between the modulations trained from the same pre-trained LM can even perform better than individual modulation (coined "model soup" [86, 93]). In this regard, we thus *aggregate* the memory bank into a single modulation based on the given input. Formally, we learn a set aggregation network $h_\psi$ that satisfies *permutation invariance* (i.e., invariance to the order of modulations in the memory bank) by utilizing cross-attention blocks [81, 36, 89] to select $\phi_i^*$:

$$\phi_i^* := h_\psi\big(g_{\theta_{\text{input}}}(\mathbf{x}_i), \{\phi_k\}_{k=1}^K\big), \tag{1}$$

where $\theta_{\text{input}}$ is the input encoder, and we use the same architectural design as the amortization network $\theta_{\text{amort}}$, albeit resorting to a reduced number of parameters for efficiency reasons. Note that $\{\phi_k\}_{k=1}^K$ is often referred to as as a context set in the meta-learning literature, hence inspiring the name of our method. We provide more architecture design details of $\theta_{\text{amort}}$ and $\psi$ in Appendix A.

**End-to-end training objective.** To learn aggregation and amortization networks, we optimize both networks in an end-to-end fashion as follows:

$$\min_{\theta_{\text{amort}}, \theta_{\text{input}}, \psi} \frac{1}{N} \sum_{i=1}^N \mathcal{L}\big(\text{LM}_{\theta_{\text{base}}}(\mathbf{x}_i; \phi_i^*), \mathbf{y}_i\big). \tag{2}$$

where $\mathcal{L}$ is the loss function, i.e., negative log-likelihood of the given label $\mathbf{y}$, and $N$ is the batch size of training query inputs and labels. Here, it is important to state that we make no updates to the static LM $\theta_{\text{base}}$, which would carry the risk of catastrophic forgetting by overwriting important parameters.

**Online adaptation stage.** After training amortization and aggregation networks based on a given training set, we now consider the online adaptation scenario. Here, we consider a stream of $K^{\mathtt{test}}$ documents $\mathbf{d}_1^{\mathtt{test}}, \cdots, \mathbf{d}_{K^{\mathtt{test}}}^{\mathtt{test}}$ given to the LM in a sequential manner, where the task input $\mathbf{x}^{\mathtt{test}}$ is not accessible during adaptation. To this end, we propose to store the compact modulations into a memory bank $\mathcal{M} \coloneqq \{g_{\theta_{\mathtt{amort}}}(\mathbf{d}_k^{\mathtt{test}})\}_{k=1}^{K^{\mathtt{test}}}$ and later predict the modulation using the aggregation network to adapt the LM, i.e., $\mathrm{LM}_{\theta_{\mathtt{base}}}(\mathbf{x}^{\mathtt{test}}; \phi^*)$ where $\phi^* \coloneqq h_\psi\big(g_{\theta_{\mathtt{input}}}(\mathbf{x}^{\mathtt{test}}), \mathcal{M}\big)$.

### 3.3 Memory efficient training and inference for MAC

Due to aforementioned challenges, the training of MAC can quickly become prohibitive. The following sections cover techniques to drastically reduce memory requirements.

**Backpropagation dropout.** During the online adaptation stage, the aggregation network is required to predict the modulation based on the memory bank, which may consist of large numbers of modulations (examples extracted from thousands of novel documents in our experimental setup). To handle large batch inference, it is crucial to present similar examples during training to avoid distribution shift between training and online adaptation stage and ensure that memory selection is robust. To this end, we propose a memory-efficient way to increase the training context size $K$ by computing gradients using only a subset of randomly chosen examples (ensuring unbiased gradient computation), thus allowing training with significantly larger memory sizes. More concretely, with probability $p$, we perform amortization at training time with a stop-gradient operation, i.e., $\mathtt{stopgrad}\big(g_{\theta_{\mathtt{amort}}}(\mathbf{d}_i)\big)$ where $p$ is a hyper-parameter, thus reminiscent of dropout. It is important to note that this random sub-sampling yields *unbiased approximation of the full gradient* under amortization-based meta-learning schemes [6], hence, does not hurt the overall performance.

**Hierarchical modulation aggregation.** In addition, we propose an efficient inference technique to deal with the accumulated memory bank. Let $T$ be the number of output tokens for each context and $K$ the number of amortized contexts, respectively. Then, the memory usage made by a single cross-attention layer becomes $\mathcal{O}(KT^2)$ (note that the input $\mathbf{x}$ is also mapped into $T$ tokens). This indicates the aggregation process requires a memory cost that linearly scales with the size of the memory bank.

To alleviate memory consumption, we propose hierarchical modulation aggregation that uses a divide-and-conquer strategy (see Algorithm 3). Specifically, for a given memory bank size of $K$ with $T$ tokens, we subgroup the total $KT$ tokens into $M$ tokens each, thereby having $\lceil \frac{KT}{M} \rceil$ groups ($\lceil \cdot \rceil$ is the ceil function, i.e., the smallest integer which is greater than or equal to the given input). Then, we aggregate the modulations of individual subgroups into a single output to obtain $\lceil \frac{KT}{M} \rceil$ modulations. We repeat this procedure until it outputs a single modulation. Assuming no parallelization, one can compute this process by only utilizing the memory complexity of $\mathcal{O}(MT)$ where $M$ is a hyperparameter (more details of the complexity calculation is in Appendix A.2).

## 4 Experiments

In this section, we provide an empirical evaluation of MAC, systematically verifying claims made throughout the manuscript and thus supporting the suitability of its constituent components. Specifically, we investigate the following questions:

- How does MAC perform compare to other online learning techniques for LMs? (Table 1 & Table 2)

- Is MAC more efficient compared to online finetuning schemes? (Figure 2)

- Does MAC show effective knowledge retention compared to other finetuning methods? (Figure 3)

- Does proposed efficient training and inference schemes save memory usage? (Figure 4 & Figure 5)

Before answering each question, we outline the experimental protocol (more details in Appendix A).

**Datasets.** For the experiment, we utilize three question-and-answering (QA) datasets including StreamingQA [45], SQuAD [62], and ArchivalQA [82], by following the prior work [26]. Here, unlike the original use of SQuAD and ArchivalQA (i.e., used for evaluating static LMs), we use these datasets for online adaptation (i.e., adapting on a stream of documents), hence, denote with an additional "-Seq" notation throughout the section.

Table 1: Comparison of the online adaptation performance between MAC and online finetuning baselines. We report the exact match (EM) and F1 score by adapting the LM on a stream of documents and then performing QA based on the learned data. $^*$ denotes the adaptation results of CaMeLS using a proxy token weighting LM (i.e., a smaller LM than the base LM) due to memory consumption, and OOM denotes unavailable results due to the running out-of-memory on a single NVIDIA A100 80GB GPU (even with a batch size of 1). The bold indicates the best result within the group.

| Model (# params) | Method | StreamingQA | | SQuAD-Seq | | ArchivalQA-Seq | |
| --- | --- | --- | --- | --- | --- | --- | --- |
| | | EM ($\uparrow$) | F1 ($\uparrow$) | EM ($\uparrow$) | F1 ($\uparrow$) | EM ($\uparrow$) | F1 ($\uparrow$) |
| DistilGPT2 (82M) | Uniform | 1.62 | 3.76 | 1.24 | 2.54 | 4.86 | 4.08 |
| | Salient Spans | 1.44 | 4.67 | 1.03 | 2.47 | 4.52 | 3.76 |
| | CaMeLS | 1.62 | 5.79 | 1.47 | 3.08 | 4.62 | 6.19 |
| | **MAC (ours)** | **5.59** | **10.18** | **2.01** | **6.85** | **7.55** | **10.58** |
| GPT2-Large (774M) | Uniform | 4.74 | 7.00 | 3.64 | 4.97 | 7.66 | 8.71 |
| | Salient Spans | 4.86 | 8.54 | 4.03 | 6.48 | 9.75 | 11.19 |
| | CaMeLS$^*$ | 5.35 | 10.60 | 4.97 | 8.63 | 9.92 | 12.41 |
| | **MAC (ours)** | **7.25** | **13.31** | **6.43** | **11.42** | **11.84** | **15.26** |
| GPT2-XL (1.5B) | Uniform | 5.11 | 7.48 | 6.10 | 6.78 | 8.61 | 10.78 |
| | Salient Spans | 5.40 | 9.42 | 4.55 | 6.74 | 11.81 | 14.11 |
| | CaMeLS$^*$ | 6.55 | 11.67 | 6.70 | 10.15 | 13.87 | 15.74 |
| | **MAC (ours)** | **8.99** | **15.38** | **7.10** | **12.55** | **14.01** | **17.12** |
| LLaMA-2 (7B) | Uniform | 12.43 | 13.54 | 13.25 | 17.01 | 18.53 | 21.35 |
| | Salient Spans | 13.33 | 18.97 | 13.74 | 18.66 | 18.97 | 22.75 |
| | CaMeLS | | | ——— OOM ——— | | | |
| | **MAC (ours)** | **14.29** | **21.79** | **15.07** | **21.14** | **20.12** | **23.90** |

**Online adaptation setup.** After training MAC (i.e., learning $\theta_{\mathtt{amort}}$, $\theta_{\mathtt{input}}$, and $\psi$ parameters) on a training dataset that consists of document and QA pairs, we evaluate the online adaptation performance on the stream of documents. Here, we use 1,665 documents to adapt the LM and then perform the evaluation after the adaptation, where QA pairs are sampled from the learned documents. Each document can consist of tokens up to 512 when using the Byte Pair Encoding [74].

**Baselines.** We mainly consider the online finetuning baselines introduced in [26], including *Uniform*, *Salient Spans* and *CaMeLS*. Here, all baselines are first pre-trained on a QA-paired training set (without the documentation) and then utilize auto-regressive finetuning to adapt to the stream of documents. Specifically, Uniform uses uniform token weighting, Salient Spans assigns uniform weight to tokens in salient spans [21] and no weights to other tokens, and CaMeLS utilizes the output of the token weighting LM (which is meta-learned to predict the important token so that the performance of the adapted LM is maximized). Furthermore, we also consider the joint usage of MAC with the retrieval augmentation scheme, including BM25 [66], Contriever [29], and DPR [33].

## 4.1 Online adaptation with MAC

We first present the main result by comparing the online adaptation performance with other baselines. Here, we mainly compare with online finetuning schemes and additionally show that MAC can be jointly used with a retrieval augmentation method to further improve the performance.

**Comparison with online finetuning methods.** In Table 1, we show the online adaptation performance of MAC and the online finetuning baselines. Overall, MAC significantly outperforms all the prior online finetuning methods by a large margin, leading to a better exact match (EM) and F1 score. We also found that CaMeLS [26] suffers from the memory shortage on LLaMA-2 even when using the memory efficient techniques (e.g., 4bit quantization [13] and ZeRO [61]), as it requires second-order gradient computation for meta-learning. Consequently, it requires a proxy model (a small-sized LM compared to the base LM) that uses the same tokenization (e.g., we use DistilGPT2 for GPT family as suggested in [26]).

Furthermore, it is worth mentioning that MAC is significantly efficient in both memory and adaptation time compared to other online finetuning methods; we remark that MAC does not require any gradient computation to update the model, while online finetuning needs the gradient to update the model. For

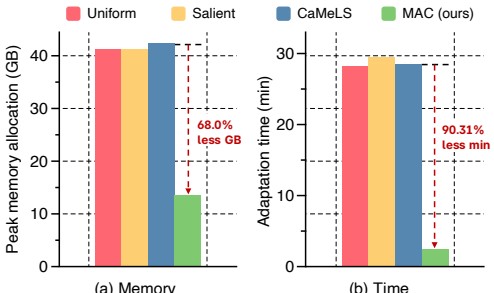

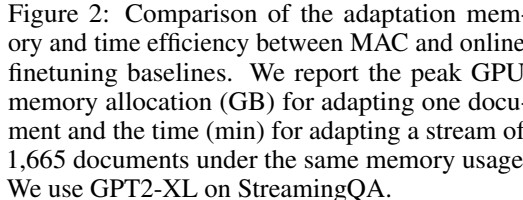

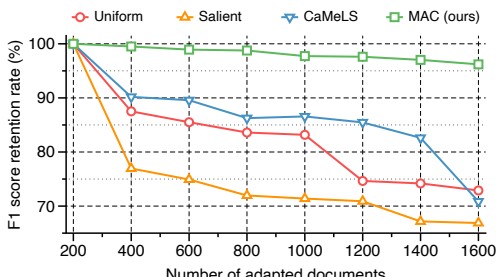

Figure 2: Comparison of the adaptation memory and time efficiency between MAC and online finetuning baselines. We report the peak GPU memory allocation (GB) for adapting one document and the time (min) for adapting a stream of 1,665 documents under the same memory usage. We use GPT2-XL on StreamingQA.

Figure 3: Catastrophic forgetting analysis under GPT2-XL trained on StreamingQA dataset. We report the F1 score retention rate (%) through measurement of relative F1 score decline in the initially adapted 200 documents during subsequent adaptation to a new stream of documents (up to additional 1,400 documents).

Table 2: Online adaptation performance of MAC jointly using the retrieval augmentation under ArchivalQA-Seq dataset. We consider BM25, Contriever, and DPR as retrieval augmentation methods. We report the exact match (EM) and F1 score by adapting the LLaMA2-7B on a stream of documents and then performing QA based on the learned data while retrieval augmentation retrieves documents. The bold indicates the best results within the group.

|  | Top-1 | | Top-3 | | Top-5 | |
|---|---|---|---|---|---|---|
|  | EM | F1 | EM | F1 | EM | F1 |
| BM25 | 48.53 | 54.17 | 56.18 | 63.74 | 64.74 | 71.83 |
| **BM25 + MAC (ours)** | **52.81** | **56.55** | **60.22** | **66.82** | **68.85** | **74.89** |
| Contriever | 44.78 | 51.55 | 52.56 | 61.28 | 60.10 | 67.83 |
| **Contriever + MAC (ours)** | **47.99** | **53.23** | **53.92** | **63.75** | **61.28** | **70.01** |
| DPR | 48.98 | 55.01 | 57.02 | 64.27 | 65.07 | 72.24 |
| **DPR + MAC (ours)** | **49.57** | **55.98** | **60.19** | **67.05** | **68.52** | **75.00** |

instance, compared to CaMeLS, MAC reduces 68.0% memory usage for a single document adaptation and can adapt 128 times larger number of documents when using the same memory. Moreover, the adaptation time reduces from 28.58 to 2.5 minutes under the same memory usage (i.e., 90.31% drop). We emphasize that both types of efficiency are crucial for online learning LMs as i) the document corpus is expanding rapidly, and ii) it enables the user to use a larger model for better generalization.

**Knowledge Retention of MAC.** We now address one of our primary motivations for this study: a comparison of knowledge retention by analyzing the catastrophic forgetting of each method. To this end, we evaluate the F1 score retention ratio, which is determined by the decline in the F1 score of the initially adapted 200 documents during the optimization on a subsequent stream of documents. As shown in Figure 3, MAC shows a strong knowledge retention compared to other online finetuning methods: when adapting additional 1,400 documents, MAC retains the initial performance by 96.2% while CaMeLS retains 70.8%. These results indeed highlight i) the benefit of using a memory bank as a tool for preserving knowledge and ii) our aggregation mechanism well predicts the modulation even when the memory bank's cardinality increases throughout the adaptation process. It is also worth noting that online finetuning schemes somewhat suffer from preserving the newly learned knowledge, especially when the number of adapted documents increases, thus may limit the practical usage for real-world applications.

**Improving MAC with retrieval augmentation.** In addition, we show that MAC can be further improved by using retrieval augmentations. Here, we note that the user requires more inference costs to use retrieval augmentations as prepending the retrieved document in front of the question quadratically increases the inference computation based on the document length due to the Attention mechanism [81]. For the experimental setup, we compare it with LMs that are pre-trained on QA training set with an appended top-1, top-3, and top-5 retrieved document for each question, i.e.,

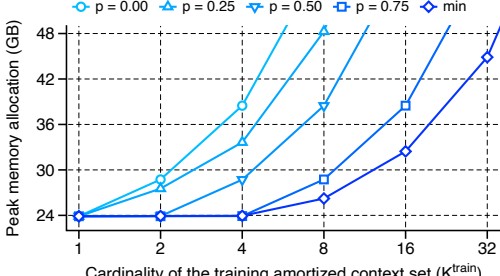
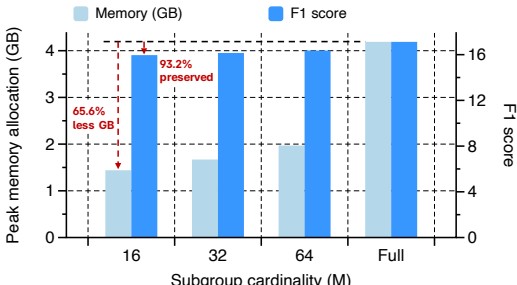

Figure 4: Memory efficiency of the backpropagation dropout. We report the peak GPU memory allocation (GB) when training GPT2-XL on StreamingQA dataset under varying sizes of amortized contexts set size ($K^{\text{train}}$). $p$ indicates the dropout ratio and 'min' denotes the full dropout except for the single document.

Figure 5: Memory efficiency of the hierarchical modulation aggregation. We report the peak GPU memory allocation (GB) and F1 score under GPT2-XL trained on ArchivalQA-Seq dataset by varying the subgroup cardinality $M$. The "Full" indicates the use of the full context set (i.e., no hierarchical aggregation).

$\text{LM}_{\theta_{\text{base}}}(\mathbf{d} \oplus \mathbf{x}; \phi)$ where $\oplus$ and $\phi$ indicate concatenation and the modulation, respectively. Here, we consider three types of popular retrieval augmentation methods, including BM25 [66], Contriever [29], and DPR [33]. As shown in Table 2, using BM25 with MAC significantly improves the performance by a large margin in all cases, e.g., F1 score of 71.83% → 74.89% for LLaMA-2 (7B) when using top-5 documents. We conjecture that the aggregation process of MAC enables the utilization of the shared information across the documents, thus improving the performance over the single document retrieval. We believe further extending MAC for the joint usage with retrieval augmentation schemes will be an interesting future direction to explore where one can extend the amortization and input network to enhance the aggregation of modulations but also learn to well retrieve documents.

## 4.2 Efficiency of backpropagation dropout and hierarchical modulation aggregation

We verify the proposed memory efficient techniques, namely the backpropagation dropout and the hierarchical modulation aggregation for training and inference, respectively. Here, we report the peak GPU utilization when using the proposed techniques to show the memory efficiency. Furthermore, we re-emphasize that such techniques are important for (i) scaling LMs to larger models and (ii) handling a large number of documents during online adaptation, which are both necessary for scaling.

**Training memory efficiency.** To show the memory efficiency of the backpropagation dropout, we increase the number of amortized contexts $K^{\text{train}}$ during training time and vary the dropout ratio $p$. As shown in Figure 4, increasing the dropout ratio can significantly handle more contexts under the same memory constraint. As a result, we found that simply using $p = 0.75$ is an

Table 3: Effect of backpropagation dropout (backprop.) on LLaMA2-7B under StreamingQA dataset. $K$ indicates the batch size.

| Method | $K$ | Memory (GB) | F1 |
|---|---|---|---|
| No backprop. | 1 | 33.86 | 12.43 |
| MAC | 4 | 34.01 | 21.79 |

effective choice when using large models (# parameters > 1B) as the training context size is small in such cases. For instance, when training LLaMA-2 (7B) model on StreamingQA dataset without this technique, one can only compute the loss with a single document (under 32 GB GPU), thus the aggregation network cannot learn the similarity between the modulations. As a result, using backpropagation dropout improves the performance of LLMs (in Table 3).

**Inference memory efficiency.** Here, we show that the hierarchical modulation aggregation can significantly reduce memory usage while effectively preserving the performance for the inference. To this end, we vary the cardinality of the subgroup $M$ and report the peak GPU memory usage and F1 score where we only measure the used memory by the modulation aggregation (i.e., excluding the LM cost). As shown in Figure 5, using the subgroup size of $M = 16$ can reduce the memory by 65.6% while still preserving 93.2% of the original accuracy. We remark that this technique can be applied even without additional training trick or regularization, demonstrating similar observations from the prior works that uses hierarchical aggregation (or merging) in the context of Transformers [5, 76], yet MAC is the first to aggregate the modulations.

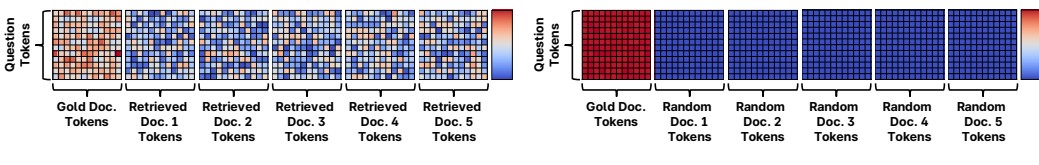

(a) BM25 retrieved documents                    (b) Random documents

Figure 6: Visualization of the per-token final layer cross-attention. The aggregation network is provided with the gold document (containing the answer) with five additional documents, which are either (a) retrieved using BM25 or (b) randomly sampled. Each question and document are encoded into $K = 12$ tokens, where $K$ is a hyperparameter. Red denotes the high similarity with the question.

## 4.3 Additional analysis

In this section, we provide more analysis of MAC. Here, we mainly consider baselines that show effectiveness in the main experiment (e.g., CaMeLS in Table 1) and consider GPT2 family trained with StreamingQA dataset.

**Cross-attention analysis.** We analyze whether the learned cross-attention is attending to the correct information. To this end, we visualize the final cross-attention layer of the aggregation network trained on StreamingQA with GPT2-Large, where we provide the gold document (containing the answer to the question) and an additional five documents. Here, we consider providing the retrieved documents using BM25 or random documents, where we average the cross-attention over 25 questions (as considering more number of questions over-smooth the visualization). As shown in Figure 6, the model selectively attends to the gold document when provided with irrelevant random documents, effectively ignoring them, while appropriately attending to relevant documents retrieved using BM25, indicating a well-trained attention mechanism capable of discerning useful information.

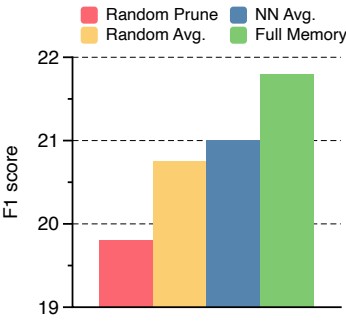

Figure 7: Comparison of various memory bank reduction methods on LLaMA2-7B.

**Memory bank size constraint.** One possible concern of MAC is the growing size of the memory bank as the number of adapted documents increases. To this end, we have conducted an additional experiment using a fixed memory bank size for MAC. Specifically, we reduce the number of amortized contexts when it reaches the memory constraint of 1,250 (where the total number of contexts is 1665). Here, we consider three simple yet effective schemes: i) random pruning, ii) randomly averaging two modulations $\phi_{\text{new}} = \frac{1}{2}(\phi_1 + \phi_2)$, and iii) averaging two nearest-neighbor (NN) modulations based on the cosine distance. As shown in Figure 7, we tested LLaMA-2 7B on StreamingQA by reducing the memory bank size where averaging NN modulations shows quite effective preservation. We believe it would be an interesting future direction to further explore MAC under memory bank size constraints where a great variety of techniques can be developed in this direction, for instance, using neural compression techniques to reduce the memory bank size [3, 73].

**Using other types of PEFT.** Here, we show that other types of PEFT modulation can also be used for our framework. To this end, we considered LoRA [25] as an alternative to P-tuning v2 [47]. As shown in Table 4, LoRA also performs well compared to other online fine-tuning methods, but overall, P-tuning v2 outperformed LoRA when training GPT2-XL on the StreamingQA dataset. This result aligns with the finding from previous work [58], where they also observed that P-tuning v2 outperforms LoRA when using amortization. Additionally, we believe P-tuning is also easy to implement, as it allows efficient batch computation, enabling a single forward pass of the LLM with different modulations. In contrast, LoRA requires separate forward passes for each modulation, which increases the training time.

Table 4: Online adaptation performance on different types of PEFT, including LoRA and P-tuning-v2. We train GPT2-XL on StreamingQA.

| PEFT type | EM | F1 |
|---|---|---|
| LoRA | 8.67 | 15.15 |
| P-tuning v2 | **8.99** | **15.38** |

**Adaptation on out-of-distribution (OOD) datasets.** We additionally analyze the online adaptation performance of MAC on the OOD dataset from the training distribution. To this end, we compare the performance with CaMeLS [26] on GPT2-XL, as other online finetuning methods do not involve a training stage (i.e., no training distribution). Here, we use StreamingQA as a training set (i.e., a relatively large dataset) and other datasets as OOD. As shown in Table 5, MAC outperforms CaMeLS in F1 score. It is worth noting that the meta-learning performance scales as the training distribution is more diverse [91], hence, we believe training MAC on larger datasets will further improve the OOD generalization.

Table 5: Online adaptation performance on OOD datasets: We report the F1 score of GPT2-XL trained on StreamingQA, adapting to SQuAD and ArchivalQA.

| StreamQA $\rightarrow$ | SQuAD | ArchivalQA |
|---|---|---|
| CaMeLS | 8.63 | 13.43 |
| **MAC (ours)** | **10.47** | **13.73** |

**Language modeling with MAC.** While the conventional evaluation protocol for online learning LMs uses QA [32, 31, 26], we additionally conducted a language modeling task (i.e., predicting the next token). Specifically, we adapted the LLM on a stream of documents, then gave the initial 10% of the document as input to the input network (this is equivalent to a question in the QA task). Here, we measured the perplexity of the remaining 90% of the documents on two cases: (i) the documents used for LLM adaptation to measure knowledge preservation and (ii) unseen documents to measure generalization. As shown in Table 6, MAC outperforms other online finetuning baselines in both cases.

Table 6: Perplexity on adapted and unseen documents. We use GPT2-Large auto-regressively trained on StreamingQA documents.

| | Adapted | Unseen |
|---|---|---|
| Uniform | 11.43 | 13.89 |
| Salient Spans | 27.87 | 29.69 |
| CaMeLS | 11.31 | 14.77 |
| **MAC (ours)** | **10.91** | **12.71** |

**Design choice for the amortization network.** Here, we consider different types of design choice for the amortization network. To this end, we evaluated three architectural configurations: decoder-only, encoder-only, and encoder-decoder language models. Specifically, we experimented with (i) the GPT2 model and (ii) the T5 encoder with learnable tokens, where input context is compacted into these tokens. As shown in Table 7, the encoder-decoder model demonstrated superior performance over other configurations, using GPT2-XL as the base LLM on the StreamingQA dataset.

Table 7: Online adaptation performance across design choices for the amortization network, evaluated by training GPT2-XL on the StreamingQA dataset.

| | EM | F1 |
|---|---|---|
| Encoder only (T5-encoder) | 8.53 | 15.01 |
| Decoder only (GPT2) | 8.01 | 14.87 |
| Encoder-Decoder (T5) | **8.99** | **15.38** |

## 5 Discussion and Conclusion

We propose MAC, an efficient and effective online adaptation framework for static LMs with strong knowledge retention. MAC compresses the context document into parameter-efficient finetuning modulations, predicted by a meta-learned amortization network. These contexts are stored in a memory bank for strong knowledge retention and aggregated into a single output when a question is input. MAC excels in performance, adaptation time, and memory efficiency, and shows superior knowledge retention for newly learned documents when handling a stream of documents.

**Future works and limitations.** We believe it will be an interesting future work extending MAC to multiple applications that require online learning in an efficient manner, e.g., federated learning for LMs [8] and model editing [52, 53, 23]. Moreover, one possible limitation of MAC is the increasing size of the memory bank during online adaptation. In this paper, we found that the memory bank can be effectively reduced by averaging nearest neighbor modulation (in Section 4.3), where we believe further investigating a better-merging technique will be an interesting future direction to explore.

**Societal impact.** This paper presents a method that enhances the online adaptation performance of LMs through the use of amortization-based meta-learning and the memory bank. Similar to other works, using memory banks for LMs in real-world applications comes with benefits and pitfalls (e.g., privacy concerns when saving documents from users), requiring the responsible use of the technology. We believe further extending the amortization network in the perspective of privacy will be an interesting future direction to explore. For instance, rather than saving the raw text as other retrieval augmentations techniques or memory-augmented LMs, one can learn to amortize the context documents to prevent the document's privacy leakage.

## Acknowledgements

We thank Nathan Hu and Minseon Kim for providing helpful feedback and suggestions in preparing an earlier version of the manuscript. This work was supported by Institute of Information & communications Technology Planning & Evaluation (IITP) grant funded by the Korea government (MSIT) (RS-2019-II190075, Artificial Intelligence Graduate School Program (KAIST), No.RS-2021-II212068, Artificial Intelligence Innovation Hub, No.2022-0-00713, Meta-learning applicable to real-world problems, and No. RS-2024-00509279, Global AI Frontier Lab) and the NIPA(National IT Industry Promotion Agency), through the Ministry of Science and ICT (Hyperscale AI flagship project).

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

# A Experimental Details

## A.1 Experimental details

**Training details.** We mainly follow the training configuration suggested by [26]. For all datasets, we train 50 epochs by using Adam [38] optimizer, where we warm up the learning rate for the first epoch (except for training DistilGPT2; 68) and then use a constant value throughout the training. Here, we use a learning rate of $1e-5$ for all models except for DistilGPT2 where it uses $1e-4$. The output token number of the amortized network $T$ is 12 for DistilGPT2 and 24 for the rest. We apply backpropagation dropout for large models with more than 1 billion parameters, using a ratio of $p = 0.75$. Additionally, we use 4bit quantization [13] and ZeRO [61] when training GPT2-XL [59], and LLaMA-2 [79] where we also (4-bit) quantize the T5 encoder [60]. It is important to note that the quantization should be applied to pre-trained networks, not the networks learned from the random initialization (e.g., amortization and aggregation network). We use a batch size of 64 for DistilGPT2 and 32 for the rest by using the gradient accumulation.

**Evaluation details.** We follow the same evaluation protocol from [26]. For the online adaptation, we adapt the model on a stream of 1,665 documents and then perform a QA evaluation. For the online finetuning baselines, we follow Hu et al. [26] to find the best learning rate hyperparameter, where we observed that the performance is somewhat quite sensitive to the choice. We mainly used $6.5e-6$ for all online finetuning methods except for CaMeLS, which uses $2.5e-5$ in most cases. For the catastrophic forgetting analysis in Figure 3, we fixed the learning rate to $6.5e-6$ for all online finetuning methods as we found that forgetting occurs more on larger learning rates. It is worth remarking that MAC does not require any additional hyperparameter during online fine-tuning.

**Base LM details.** We mainly consider GPT2 family [59] as the static base LM $\theta_{\text{base}}$ by following the prior work [26], where we additionally conduct the experiment on LLaMA-2 [79] to verify the scalability of MAC. For the amortization network, we consider the T5 model family [60] that are relatively smaller than the base LM. It is important to note that the output number of tokens $T$ of the amortization and aggregation networks is a hyper-parameter, where we use 24 for all architectures except for Distil-GPT2, which uses 12. Then, we map these $T$ tokens into each layer's modulation through a linear layer where we use P-tuning v2 [47] as the modulation design.

**Amortization network details.** For the model details, we mainly describe the design choice of our amortization $\theta_{\text{amort}}$. Note that input encoder $\theta_{\text{input}}$ uses the same architectural design as $\theta_{\text{amort}}$ while using a smaller sized network. For the amortization network, we follow the design choice from [58] and use the T5 encoder-decoder model [60] as the base architecture. Specifically, we learn trainable tokens that are used for decoder input so that the output number of tokens $T$ is consistent. Then, we have an individual two-layered MLP for each output token. For the network size, we use T5-small as the amortization $\theta_{\text{amort}}$ network for Distil-GPT2, T5-base for GPT2-Large, and T5-Large for both GPT2-XL and LLaMA-2 (7B) where the input network $\theta_{\text{input}}$ uses a smaller model (T5-small for Distil-GPT2 and T5-base for the rest).

**Aggregation network details.** The aggregation network uses four cross-attention blocks, each consisting of one cross-attention layer and one feed-forward network. Here, the set of parameter efficient finetuning (PEFT) modulations (in the memory bank) is the key and value of each cross-attention layer, and the encoded question ($g_{\theta_{\text{input}}}(\mathbf{x})$; soft prompt tokens) is the initial query of the cross attention layer (i.e., later layers use the previous block's output as the query input). Thereby, the output of the aggregation network is soft prompts that have the same dimension as the encoded question.

**Dataset details.** Here, we describe the dataset detail in the following.

○ **StreamingQA** [45] The StreamingQA is composed of questions that are either created by annotators or produced using a large-scale language model. These questions can be answered using a dynamic knowledge database of English WMT news articles, which have been timestamped and were published from 2007 to 2020, and these articles are also included in the dataset. Following the setups in [26], we use 21k training questions, 1.7k validation questions, and 5k test questions, respectively. Also, the same number of documents with the questions is used for each split, during the experiments. For the baselines that require QA pre-training (see Section 4), we use 40k training questions and 4k validation questions, respectively.

- **SQuAD** [62]: The Stanford Question Answering Dataset (SQuAD) is composed of questions created by crowdworkers based on a collection of Wikipedia articles, where the answer to each question is a span contained in the corresponding article. Following the setups in [26], we use 39.9k training questions, 5.6k validation questions, and 10.6k test questions, respectively. Next, we use 8.6k training documents, 1.2k validation documents, and 2.1k test documents, respectively. For the baselines that require QA pre-training (see Section 4), we use 40k training questions and 2.1k validation questions, respectively.

- **ArchivalQA** [82]: The ArchivalQA dataset is constructed with synthetically generated questions from the sophisticatedly designed pipelines with language models. Specifically, questions are generated from articles in the New York Times Annotated Corpus [67]. Also, the answer to each question is a span contained in an article. Following the setups in [26], we use 21.7k training questions, 5.3k validation questions, and 8.7k test questions, respectively. Next, we use 12.8k training documents, 3.0k validation documents, and 5.0k test documents, respectively. For the baselines that require QA pre-training (see Section 4), we use 12.4k training questions and 3k validation questions, respectively.

## A.2 Memory complexity of hierarchical modulation aggregation

The calculated memory complexity is based on the Attention map size, which is equal to the dimension after multiplying the Query and Key of the Cross-Attention layer. Here, the Query dimension is fixed to $T$ tokens, and the Key dimension is dependent on the size of the memory bank. In this regard, $K$ documents are encoded into $KT$ tokens, thus showing $\mathcal{O}(KT^2)$ for the entire set aggregation. For the hierarchical aggregation, we subgroup $KT$ tokens into $M$ tokens for each memory, thus reducing the complexity into $\mathcal{O}(MT)$. Here, it is important to note that we do not assume parallelization for the hierarchical aggregation when computing each subgroup, hence, the memory complexity is $\mathcal{O}(MT)$.

# B  Algorithm

## B.1  Algorithm of MAC

---

**Algorithm 1** Meta-training of MAC

**Input:** $\theta_{\texttt{amort}}, \theta_{\texttt{input}}, \theta_{\texttt{base}}, \psi, \mathcal{C}^{\texttt{train}}$, learning rate $\beta$

1: **while** not converge **do**
2:     Sample documents $\{\mathbf{d}_1, \ldots, \mathbf{d}_K\}$ from $\mathcal{C}^{\texttt{train}}$.
3:     Sample QA pairs $(\mathbf{x}_k, \mathbf{y}_k) \sim p(\mathbf{x}, \mathbf{y}|\mathbf{d}_k)$.
4:     **for** $k = 1$ to $K$ **do**
5:         # Summarize context
6:         $\phi_k = g_{\theta_{\texttt{amort}}}(\mathbf{d}_k)$
7:     **end for**
8:     # Aggregate modulations
9:     $\phi_k^* = h_\psi\big(g_{\theta_{\texttt{input}}}(\mathbf{x}_k), \{\phi_k\}_{k=1}^K\big)$
10:    # Compute loss
11:    $\mathcal{L}_{\texttt{total}} = \mathbb{E}_k[\mathcal{L}\big(\text{LM}_{\theta_{\texttt{base}}}(\mathbf{x}_k; \phi_k^*), \mathbf{y}_k\big)]$
12:    # Optimize
13:    $\theta_{\texttt{amort}} \leftarrow \theta_{\texttt{amort}} - \beta\nabla_{\theta_{\texttt{amort}}}\mathcal{L}_{\texttt{total}}$
14:    $\theta_{\texttt{input}} \leftarrow \theta_{\texttt{input}} - \beta\nabla_{\theta_{\texttt{input}}}\mathcal{L}_{\texttt{total}}$
15:    $\psi \leftarrow \psi - \beta\nabla_\psi\mathcal{L}_{\texttt{total}}$
16: **end while**

**Output:** $\theta_{\texttt{amort}}, \theta_{\texttt{input}}, \psi$

---

**Algorithm 2** Online learning of MAC

**Input:** Stream of document $\mathcal{C}^{\texttt{test}}$, test QA set $\{\mathbf{x}_i, \mathbf{y}_i\}_{i=1}^I, \theta_{\texttt{amort}}, \theta_{\texttt{input}}, \theta_{\texttt{base}}, \psi$

1: Initialize new memory bank $\mathcal{M} := \emptyset$
2: Extract amortized contexts from the stream of documents
3: **for** $k = 1$ to $K^{\texttt{test}}$ **do**
4:    # Summarize context
5:    $\phi_k = g_{\theta_{\texttt{amort}}}(\mathbf{d}_k)$
6:    Save $\phi_k$ into $\mathcal{M}$
7: **end for**
8: Adapt the LM based on the input and evaluate
9: **for** $i = 1$ to $I$ **do**
10:    # Aggregate modulations
11:    $\phi_i^* = h_\psi\big(g_{\theta_{\texttt{input}}}(\mathbf{x}_i), \{\phi_i\}_{i=1}^{K^{\texttt{test}}}\big)$
12:    $\mathbf{y}_i^{\texttt{pred}} = \text{LM}_{\theta_{\texttt{base}}}(\mathbf{x}_i; \phi_i^*)$
13: **end for**

**Output:** $\text{Accuracy}\big(\{(\mathbf{y}_i, \mathbf{y}_i^{\texttt{pred}})\}_i^I\big)$

---

## B.2   Algorithm of the hierarchical modulation aggregation

---

**Algorithm 3** Hierarchical modulation aggregation

---

**Input:** $\mathcal{M}, \psi, \mathbf{x}, \theta_{\texttt{input}}$, subgroup cardinality $M$

1: **while** $|\mathcal{M}| > 1$ **do**
2:     Subgroup $\mathcal{M}$ into $M$ tokens $\{\mathcal{M}_1, \cdots, \mathcal{M}_{\lceil \frac{|\mathcal{M}|}{M} \rceil}\}$
3:     Initialize new memory bank $\mathcal{M}_{\texttt{new}} := \emptyset$
4:     **for** $i = 1$ to $\lceil \frac{|\mathcal{M}|}{M} \rceil$ **do**
5:         Aggregate subgroup $\phi_i \leftarrow h_\psi\big(g_{\theta_{\texttt{input}}}(\mathbf{x}), \mathcal{M}_i\big)$
6:         Store $\phi_i$ into $\mathcal{M}_{\texttt{new}}$
7:     **end for**
8:     Repeat by $\mathcal{M} \leftarrow \mathcal{M}_{\texttt{new}}$
9: **end while**

**Output:** $\mathcal{M} = \{\phi^*\}$

---

## C   More Discussion with Related Work

**Prompt compression.** The amortization meta-learning scheme of MAC can also be related to prompt compression methods [85, 12]. The major goal of prompt compression techniques is to reduce the context length while preserving the prediction performance. While seemingly similar to our amortization-based meta-learning approach (as it compresses the document into a few tokens), our amortization network learns to extract the new knowledge that is useful to adapt the base LM's old knowledge. Namely, their goals are different. Nevertheless, we believe exploring the architectures suggested in other prompt compression schemes to improve our amortization network will be an interesting future direction to explore.

## D   More Experimental Results

### D.1   Effect of train time quantization for aggregation network

Table 8: Effect of train time quantization on aggregation network. Here, we train MAC on LLaMA2 under 4bit quantization and 16bit mixed predicsion, respectively. We report exatch match (EM) and F1 score as a evaluation metric.

|  | StreamingQA | | SQuAD | | ArchivalQA | |
| --- | --- | --- | --- | --- | --- | --- |
|  | EM | F1 | EM | F1 | EM | F1 |
| 4bit quantize (nf4) | 14.29 | 21.79 | 15.07 | 21.14 | 20.12 | 23.90 |
| 16bit (bfloat16) | 19.26 | 27.20 | 16.08 | 22.34 | 21.50 | 26.25 |

We found that the main reason for the smaller improvement in larger models is due to the strong quantization applied during training, not because of our method itself. Specifically, when training large models (e.g., LLaMA4), we used 4-bit quantization for efficiency. We observed that removing this quantization (using only mixed precision training) significantly improved model performance. For example, the F1 score of Llama2 on ArchivalQA increased from 23.90% to 26.25% (as shown in the table below). This is because training with additional modules learned from scratch (e.g., aggregation network) requires careful quantization. It is worth noting that we have only removed 4-bit quantization for training, not for the adaptation stage, thereby maintaining a fair comparison with the baseline.

## D.2 Comparison with memory augmented LMs

Table 9: Comparison with memory augmented LM by compressing the context using a recent method (i.e., CCM), then learning to retrieve the relevant compressed document using a retriever. Here, we train LLaMA2 (unquantized) on StreamingQA dataset. The bold indicates the best result.

|  | EM | F1 |
|---|---|---|
| CCM + T5 encoder Retriever | 17.98 | 25.98 |
| MAC | **19.26** | **27.20** |

We also have conducted a comparison by combining the context compression method CCM [37] and RAG to show the effectiveness of MAC. Here, we first train the CCM to compress the context, then train an encoder-only model (i.e., T5 encoder) that retrieves the correct compressed contexts. For a fair comparison, we have frozen the base LLM parameter to retain the knowledge learned from the past and did not apply quantization during training. As shown in Table 9, MAC shows better performance compared to CCM combined with RAGs.

## D.3 Data contamination check for evaluation datasets

Table 10: Dataset contamination check on StreamingQA dataset by comparing document adapted performance with zero-shot and few-shot in-context learning (ICL).

| Model | Zero-shot | 5-shot ICL | Ours |
|---|---|---|---|
| GPT2-XL | 7.12 | 10.78 | 15.38 |
| LLaMA2 | 12.59 | 13.98 | 21.79 |

We measured the base LLM's zero-shot and 5-shot in-context learning (ICL) F1 accuracies on the StreamingQA dataset to verify whether the model has already learned the test set knowledge. As shown in Table 10, the base LLM struggles to answer the evaluation set without adaptation to the test set documents, indicating the low possibility of test set leakage.

