# OpenReview forum: "Online Adaptation of Language Models with a Memory of Amortized Contexts"
_NeurIPS.cc/2024/Conference — NeurIPS 2024 poster_

### Official Review · Reviewer_pwuf · 2024-07-12

**Soundness:** 3
**Presentation:** 3
**Contribution:** 3
**Rating:** 6
**Confidence:** 4

**Summary:**

This paper focuses on how to adapt static language models (LMs) with streaming documents during inference time.

There are two high-level challenges here: 1) how to store new domain/task relevant information, 2) how to utilize the stored information for downstream task-solving, i.e., doing question answering (QA).

In this paper, the authors propose, MAC, a parameter-efficient adaptor approach to 1) encode new documents into a sizable vector memory bank, and 2) utilize those encoded knowledge via extra attention mechanism.
The proposed method is compared with other fine-tuning baselines on three QA tasks tailored for evaluating online adaption scenarios, showing that the method is more effective.
Additional analyses and ablative studies are also provided to drive more insights into the model designs and behaviors.

**Strengths:**

1) The paper is well-written with enough background and details for readers to follow.

2) The authors apply their proposed method to different LMs with various architectures and training protocols, which provide support for the generalization ability of the approach.

3) The design of experiments are mostly reasonable (baselines are OK, multiple datasets are good), and the results suggest the proposed method is effective. There are additional analyses and ablative studies, e.g., i) the proposed method can do better knowledge retention and help RAG, ii) most design choices are validated.

**Weaknesses:**

1) The proposed method is a natural extension of memory-augmented LLM (as cited in paper [76,79]) with a token compression module. Rather than comparing with only closed-book QA models, it is good to compare with context compression methods, e..g, text token compression w/ RAG or long-context models, e.g.,

[1]J.-H. Kim, J. Yeom, S. Yun, and H. O. Song. Compressed context memory for online language

[2]H. Jiang, Q. Wu, X. Luo, D. Li, C.-Y. Lin, Y. Yang, and L. Qiu. Longllmlingua: Accelerating and enhancing llms in long context scenarios via prompt compression

[3]T. Ge, J. Hu, L. Wang, X. Wang, S.-Q. Chen, and F. Wei. In-context autoencoder for context compression in a large language model
model interaction, 2024

2) The experiment settings can be problematic.

As the goal is to adapt the model to new knowledge, it is not clear whether those evaluated datasets manifest that. Both news articles and wikipedia pages are highly used in pretraining LMs, e.g., LLaMA-2.

It is necessary to report zero-shot and few-shot results with the base model. Without that, it is hard to judge the benefit of online adaption. As all reported models have very low EM or F1 scores, it is good to report the base model for sanity check.

It is noticeable that the proposed method has less improvement on more capable models (larger sizes, e.g., LLaMA-2). It is good to dig a little bit into this, e.g., experiment with similar sized models ***with or without instruction tuning*** (LLAMA and Vicuna/Alpaca). Specifically, the instruction tuning might be relevant for the model to memorize certain information.

**Questions:**

Are those methods reported in Table 1 sensitive to the input order? For example, SQuAD questions do not have any temporal dependency on documents, it is good to see the performance on different streams, e.g., recency bias (putting the relevant documents in the beginning and irrelevant in the end).

What is the setup for decoding? Beam search or greedy? Using sampling techniques? If so, what is the temperature?

What is the training cost? Vs baselines?

What are the inference prompts?

How data efficient is the proposed method? Is it possible to achieve similar performance with less training?

---

> ### Author Rebuttal · Authors · 2024-08-07
>
> Dear reviewer pwuf,
>
> We sincerely appreciate your efforts and insightful comments to improve the manuscript.\
> We respond to each of your comments one-by-one in what follows.
>
> ---
>
> **[W1] Comparison with memory-augmented networks by combining context compression [1] with RAGs.**
>
> Thank you for the suggestion. We want to clarify that the context compression method and amortization-based meta-learning approach have different goals. The major goal of context compression techniques is to reduce the context length while preserving the prediction performance. While seemingly similar to our amortization-based meta-learning approach (as it compresses the document into a few tokens), our amortization network learns to extract the new knowledge that is useful for adapting the base LM’s old knowledge.
> Nevertheless, following your suggestion, we have conducted a comparison by combining the context compression method CCM [1] and RAGs. Here, we first train the CCM to compress the context, then train an encoder-only model (i.e., t5 encoder) that retrieves the correct compressed contexts. For a fair comparison, we have frozen the base LLM parameter to retain the knowledge learned from the past and did not apply quantization during training. As shown in the table below, MAC shows better performance compared to CCM combined with RAGs.
>
> \begin{array} {l cc}
> \hline
> \text{Llama2} & \text{EM} & \text{F1} \newline
> \hline
> \text{CCM} & 17.98 & 25.98 \newline
> \text{MAC} & \textbf{19.26} & \textbf{27.20} \newline
> \hline
> \end{array}
> \* Both methods did not apply quantization during training; therefore, the reported score of MAC is higher than the paper’s result (see more details in [W2-2]).
>
> ---
>
> **[W2-1] Is the adaptation dataset already seen by the language model? Performance of zero/few-shot.**
>
> We first clarify that we used the same setup from the previous paper [2], in which the authors carefully selected the datasets that are not trained by LMs.
>
> Nevertheless, we also agree with the reviewer’s concern and measured the base LLM's zero-shot and 5-shot F1 accuracies on the StreamingQA dataset to verify whether the model has already learned the test set knowledge. As shown in the table below, the base LLM struggles to answer the evaluation set without adaptation to the test set documents, indicating the low possibility of test set leakage.
>
> \begin{array} {l cc}
> \hline
> & \text{Zero-shot} & \text{5-shot} & \text{Ours} \newline
> \hline
> \text{GPT2-XL} & 7.12 & 10.78 & 15.38 \newline
> \text{Llama2} & 12.59 & 13.98 & 21.79 \newline
> \hline
> \end{array}
>
> ---
>
> **[W2-2] Less improvement on larger models.**
>
> Thank you for pointing this out. We found that the main reason for the smaller improvement in larger models is due to the strong quantization applied during training, not because of our method itself. Specifically, when training large models (e.g., Llama2), we used 4-bit quantization for efficiency. We observed that removing this quantization (using only mixed precision training) significantly improved model performance. For example, the F1 score of Llama2 on ArchivalQA increased from 23.90% to 26.25% (as shown in the table below). This is because training with additional modules learned from scratch (e.g., aggregation network) requires careful quantization. It is worth noting that we have only removed 4-bit quantization for training, not for the adaptation stage, thereby maintaining a fair comparison with the baseline.
>
> \begin{array} {l cccccc}
> \hline
> & \text{StreamingQA} & & \text{SQuAD} & & \text{ArchivalQA} & \newline
> \hline
> & \text{EM} & \text{F1} & \text{EM} & \text{F1} & \text{EM} & \text{F1} \newline
> \hline
> \text{4bit quantize (nf4)} & 14.29 & 21.79 & 15.07 & 21.14 & 20.12 & 23.90 \newline
> \text{16bit (bfloat16)} & 19.26 & 27.20 & 16.08 & 22.34 & 21.50 & 26.25 \newline
> \hline
> \end{array}
>
>
> ---
>
> **[Q1] Sensitivity to the input order.**
>
> We clarify that the input order does not change the output, as our aggregation network (i.e., cross-attention network) is permutation-invariant.
>
> ---
>
> **[Q2] Decoding setup.**
>
> We have followed the same decoding setup from CaMeLS [2], where we use beam search with 12 beans and do not perform sampling.
>
> ---
>
> **[Q3] Training cost compared to baselines.**
>
> Thank you for pointing this out. MAC is highly efficient compared to the major baseline CaMeLS. For instance, MAC is 32 times efficient in terms of the memory usage on the same sized model (CaMeLS requires a 80GB GPU memory to train DistilGPT (82M) with a batch size of 1 while MAC can train a batch size of 32), thus showing more than 20 time faster training speed.
>
> ---
>
> **[Q4] Inference prompts.**
>
> We do not have a specific inferent prompt or prompt template. We give the raw question to the base model (e.g., GPT2) and the raw context to the amortization network.
>
> ---
>
> **[Q5] Data efficiency of the proposed method.**
>
> Thank you for the interesting question. We have trained MAC using 21,000 document question pairs for StreamingQA (in the paper), where we reduced the document by 20% and 50% to measure the data efficiency. Here, we found that MAC is somewhat data-efficient: removing 20% of the data still shows a good performance, achieving 21.01% of F1 score in StreamingQA, where the original F1 score is 21.79% where removing more than 50% of the dataset can drop the performance to 19.75%. We believe diverse and complex document sets indeed help to train the aggregation network to better aggregate the optimal modulation.
>
> ---
>
> **Reference**\
> [1] Compressed Context Memory For Online Language Model Interaction, ICLR 2024\
> [2] Meta-Learning Online Adaptation of Language Models, EMNLP 2023

---

> > ### Comment · Reviewer_pwuf · 2024-08-07
> >
> > Thanks for the effort in responding to my questions.
> > It resolves most of my conerns.
> > I do not have any further comments.

---

> > > ### Author Response · Authors · 2024-08-07
> > > **Thank you very much for the response**
> > >
> > > Dear reviewer pwuf,
> > >
> > > Thank you very much for letting us know! We are happy to hear that our rebuttal addressed your questions well.\
> > > Also we thank you for your prompt response.
> > >
> > > If you have any further questions or suggestions, please do not hesitate to let us know.
> > >
> > > Thank you very much,\
> > > Authors

---

### Official Review · Reviewer_QtU4 · 2024-07-12

**Soundness:** 3
**Presentation:** 3
**Contribution:** 3
**Rating:** 5
**Confidence:** 4

**Summary:**

This paper proposes Memory of Amortized Contexts (MAC) which can encode the documents into compact modulations stored in a memory bank, which can later be retrieved to answer questions.

**Strengths:**

1. The proposed method is efficient compared to the baselines.
2. The paper is well-written and easy to follow.

**Weaknesses:**

1. **Unfair Comparison**: This paper only compares with [1], which may be somewhat unfair. In the paper [1], they propose to distill the information into a parameter vector $\phi$. However, in the current paper, the proposed method MAC stores all the modulations constructed from each context, which naturally contains all the knowledge from the contexts, leading to better knowledge retention. The setting might be unfair.
2. **Missing Baselines**: What I think could be the fair comparisons might be comparing MAC with most recent retrieval methods, such as DPR[2], BM25, RAPTOR[3], IRCoT[4] etc. Although the paper proposes to combine MAC and BM25, there is no direct comparison between them as MAC is also essentially doing retrieval. Having two retrievers is better than only having one retriever may not show the effectiveness of MAC.
3. **Doubt on the Performances**: Even MAC shows improvements over [1] (in the paper's setting), the best performance of MAC on Llama2-7B, ArchivalQa-Seq is 20.12 (EM), while BM25 can easily achieve 52.81 (EM) as shown in Table 2. It seems that MAC performs much worse than BM25. Can I expect the more advanced RAG methods can easily beat MAC even equipped with BM25? Can I really expect there is going to be improvements over other strong RAG methods when equipped with MAC? These are unanswered questions.
4. **Missing Related Works**: The paper talked about Retrieval Augmentation for LMs and Memory augmented LMs. However, for RAG, [2][3][4] are not mentioned in related work (there are more RAG method out there); for Memory augmented LMs, some methods such as MemoryLLM [5], Memoria [6], MemLLM [8], MemoryBank[7], Camelot[8] are not mentioned which can also perform online learning. Maybe the memory-based methods should also be used for comparison in the experiments.


[1] Meta-Learning Online Adaptation of Language Models.
[2] Dense Passage Retrieval for Open-Domain Question Answering.
[3] RAPTOR: Recursive Abstractive Processing for Tree-Organized Retrieval.
[4] Interleaving Retrieval with Chain-of-Thought Reasoning for Knowledge-Intensive Multi-Step Questions.
[5] MemoryLLM: Towards Self-Updatable Large Language Models.
[6] Memoria: Resolving Fateful Forgetting Problem through Human-Inspired Memory Architecture.
[7] MemoryBank: Enhancing Large Language Models with Long-Term Memory.
[8] MemLLM: Finetuning LLMs to Use An Explicit Read-Write Memory.
[9] CAMELoT: Towards Large Language Models with Training-Free Consolidated Associative Memory.

**Questions:**

See the weaknesses part

**Limitations:**

yes the authors have addressed the limitations

---

> ### Author Rebuttal · Authors · 2024-08-07
>
> Dear reviewer QtU4,
>
> We sincerely appreciate your efforts and insightful comments to improve the manuscript.\
> We respond to each of your comments one-by-one in what follows.
>
> ---
>
> **[W1] Unfair comparison: Online learning distils information into parameter vector where MAC stores the modulation.**
>
> We would like to clarify that distilling (or compressing) the updated information into PEFT parameters rather than the full parameter space is the key novelty of our framework, which prevents the model from forgetting the learned knowledge. While we agree with the reviewer's point that MAC increases the overall parameter count by storing such parameters in the memory bank, we emphasize that MAC outperforms larger models of baselines with a smaller-sized model, thus showing parameter efficiency. For instance, we achieved a 13.31% F1 score with GPT-2 Large (774 M model parameters + 26 M memory bank parameters = 800 M in total), whereas CaMeLS reached 11.67% with GPT-2 XL (1.5B model parameters). Moreover, as illustrated in Figure 6, we have proposed an effective method to constrain the size of the memory bank (i.e., averaging similar modulations to reduce the memory), preventing the network from increasing its parameters during adaptation. In this regard, we believe the comparison is fair, as we suggested an alternative of online finetuning based on the same training/evaluation setup.
>
> ---
>
> **[W2&W3] More comparison with recent RAGs/Can joint usage of MAC and recent RAGs consistently improve the performance?**
>
> Following your suggestion, we considered two advanced and commonly used RAG methods: DPR [1] and Contriever [2]. In our experiments, we found that BM25 remains a strong baseline, demonstrating performance comparable to RAGs in our setup. This is consistent with other literature highlighting BM25 as an effective baseline [3].
>
> More importantly, we observed that the combined use of MAC and advanced RAGs consistently yields improvements, suggesting that the benefits from RAG and MAC are orthogonal. While RAGs are effective at capturing details from retrieved documents, they heavily rely on retrieval accuracy, which can be problematic if the wrong documents are retrieved. In contrast, MAC can attend to multiple documents simultaneously using an aggregation network, allowing the LLM to capture shared information across documents. Therefore, we believe MAC and RAG complement each other well to improve the performance.
>
> \begin{array}{lcccccc}
> \hline
> & \text{Top-1} & & \text{Top-3} & & \text{Top-5} & \newline
> \hline
> & \text{EM} & \text{F1} & \text{EM} & \text{F1} & \text{EM} & \text{F1} \newline
> \hline
> \text{BM25} & 48.53 & 54.17 & 56.18 & 63.74 & 64.74 & 71.83 \newline
> \text{BM25+MAC} & \textbf{52.81} & \textbf{56.55} & \textbf{60.22} & \textbf{66.82} & \textbf{68.85} & \textbf{74.89} \newline
> \hline
> \text{Contriever} & 44.78 & 51.55 & 52.56 & 61.28 & 60.10 & 67.83 \newline
> \text{Contriever + MAC} & \textbf{47.99} & \textbf{53.23} & \textbf{53.92} & \textbf{63.75} & \textbf{61.28} & \textbf{70.01} \newline
> \hline
> \text{DPR} & 48.98 & 55.01 & 57.02 & 64.27 & 65.07 & 72.24 \newline
> \text{DPR + MAC} & \textbf{49.57} & \textbf{55.98} & \textbf{60.19} & \textbf{67.05} & \textbf{68.52} & \textbf{75.00} \newline
> \hline
> \end{array}
> Lastly, we would like to highlight the inference efficiency of our method. While RAG requires appending retrieved documents to the context, which increases the inference cost, MAC adapts the model with PEFT modulation, thus maintaining the base LLM's inference cost. Note that we only use a prefix size of 2 for each layer as the PEFT modulation, thus enabling efficient inference. As shown in Figure 1, combining MAC with RAG minimally increases memory utilization while significantly enhancing performance.
>
> ---
>
> **[W4] More related works.**
>
> We thank the reviewer for the suggestion. In the revised paper, we will include all the references pointed out by the reviewer and discuss their relevance and differences. For instance, while MAC can be categorized as a memory-augmented system, it is particularly specialized in online learning. Our core idea is to avoid updating the parameters of the base LLM to preserve the knowledge obtained from extensive pre-training while effectively updating the knowledge through the memory. In contrast, other memory-augmented networks require architectural modifications to incorporate memory, which can lead to the potential loss of pre-trained knowledge. Therefore, our approach maintains the integrity of the pre-trained model while enabling efficient online learning.
>
> ---
>
> **Reference**\
> [1] Dense Passage Retrieval for Open-Domain Question Answering, EMNLP 2020\
> [2] Unsupervised Dense Information Retrieval with Contrastive Learning, TMLR 2022\
> [3] Improving Passage Retrieval with Zero-Shot Question Generation, ACL 2022\
> [4] Compressed Context Memory For Online Language Model Interaction, ICLR 2024

---

> > ### Comment · Area_Chair_voae · 2024-08-13
> > **Please respond to the responses from the authors**
> >
> > Dear reviewer QtU4
> >
> > Could you please take a look at the responses of the authors and let us know your thoughts on them? Are you satisfied with the responses and do you have some updates on your comments?
> >
> > AC

---

> > > ### Comment · Reviewer_QtU4 · 2024-08-13
> > > **Response to the rebuttal**
> > >
> > > Thank the authors for the detailed rebuttal.
> > >
> > > The first concern was addressed, I agree that smaller model with a memory bank outperforming larger model can demonstrate the point. The second concern was that MAC cannot outperform RAG methods itself, it is also not entirely orthogonal to all RAG methods. Thus even MAC can help with other RAG methods, it is more like the combination of two RAG methods.
> > >
> > > Given these points, I have raised my score.

---

> > > > ### Author Response · Authors · 2024-08-14
> > > > **Thank you very much for the response**
> > > >
> > > > Dear Reviewer QtU4,
> > > >
> > > > Thank you very much for letting us know! We are delighted to hear that our rebuttal addressed your questions well.\
> > > > Due to your valuable and constructive suggestions, we do believe that our paper is much improved.\
> > > > Also, thank you for the overall positive review in our paper.
> > > >
> > > > In the final draft, we will add an in-depth discussion of the similarities and differences between RAG and MAC. We also believe MAC and RAG has similarities as both methods store the knowledge and utilize them base on the user query, while the main difference is that MAC attend to multiple documents simultaneously using the aggregation network, allowing the LLM to capture shared information across documents. We thus believe that the joint usage benefits RAG, as MAC can guide RAG to capture missing information not retrieved by the retriever.
> > > >
> > > > We thank the reviewer again for reviewing our paper, and giving suggestions to improve our manuscript.
> > > >
> > > > Thank you very much,\
> > > > Authors

---

### Official Review · Reviewer_j5pf · 2024-07-13

**Soundness:** 2
**Presentation:** 3
**Contribution:** 3
**Rating:** 5
**Confidence:** 4

**Summary:**

The paper proposes an online learning framework called MAC (Memory of Amortized Contexts) designed to efficiently adapt large language models (LLMs) online. By using feature extraction and memory augmentation methods, MAC compresses and stores new document information in a memory bank, retrieving relevant knowledge when answering questions. This method utilizes amortization-based meta-learning, achieving efficient modulation learning through a single forward pass, avoiding the need for gradient updates during testing. Experiments demonstrate that MAC outperforms existing methods in online adaptation performance, time, and memory efficiency, and can be combined with popular alternatives like Retrieval-Augmented Generation (RAG) to further enhance performance.

**Strengths:**

1. The paper presents a novel memory-augmented online adaptation framework based on amortization, which can efficiently adapt to new information without requiring gradient updates.

2. Experimental results show that MAC performs exceptionally well across multiple datasets and architectures, significantly improving online adaptation performance, time, and memory efficiency.

3. MAC can be combined with existing methods like RAG, enhancing the quality of retrieved documents, demonstrating good scalability and compatibility.

4. The use of two memory-efficient techniques during training and inference stages reduces memory requirements, ensuring the method's scalability.

**Weaknesses:**

1. The method involves several complex steps and model components, such as the amortization network and aggregation network, which may increase implementation difficulty.

2. Although the method has been experimentally evaluated on multiple datasets, the tasks are primarily focused on question-answering. Its adaptability to other task types remains to be verified.

3. The paper mainly demonstrates the method's effectiveness through experimental results but lacks in-depth theoretical analysis of certain key design choices, particularly in amortization and aggregation strategies.

**Questions:**

1. How is the storage overhead of MAC controlled when facing large-scale data streams? Are there any further optimization measures?
2. How does the method perform on other task types (e.g., text classification, generation tasks)? Have any related experimental evaluations been conducted?
3. How were the architectures of the amortization and aggregation networks determined? Have other architectures been tried, and what were the outcomes?
4. Is there a concern of outdated or redundant modulation parameters in the memory bank? How are these issues addressed to maintain model efficiency?

**Limitations:**

yes

---

> ### Author Rebuttal · Authors · 2024-08-07
>
> Dear reviewer j5pf,
>
> We sincerely appreciate your efforts and insightful comments to improve the manuscript. We respond to each of your comments one-by-one in what follows.
>
> ---
>
>
>
> **[W1] Possible implementation difficulty due to a somewhat complex method.**
>
> We respectfully argue that MAC is a simple framework requiring the implementation of only two networks: the amortization and aggregation networks, both of which only need a simple modification of existing code. Moreover, we have provided the PyTorch implementation of MAC in the supplementary material, and we plan to open-source the code if the paper is accepted.
>
> ---
>
> **[W2&Q2] Focused on QA: other tasks need to be verified.**
>
> We carefully remark that we already considered an additional scenario (i.e., the language modeling) rather than QA in Table 5, where we outperformed other online finetuning baselines. For your convenience, we have presented the table below. Specifically, we adapt the LLM on a stream of documents, then give the initial 10% of the document as input and measure the perplexity of the remaining documents (the initial document is equivalent to a question in the QA task). Here, we measure the perplexity of the predicted text on two document sets: i) the documents used for LLM adaptation to measure knowledge preservation and ii) unseen documents to measure the generalization, where MAC has outperformed the baselines in both cases.
>
> \begin{array} {lcc}\newline \hline &\text{Adapted documents} &\text{Unseen documents} \newline \hline \text{Uniform} & 11.43 & 13.89 \newline \text{SSM} & 27.87 & 29.69 \newline \text{CaMeLS} & 11.31 & 14.77 \newline \text{MAC (Ours)} & \textbf{10.91} & \textbf{12.71} \newline \hline\newline \end{array}
>
> Furthermore, we clarify that the major reason we mainly focused on the QA task is that it is a conventional evaluation protocol for online learning LMs [1,2,3], as evaluating the updated (or preserved) knowledge is non-trivial for other tasks. In this regard, we followed the same experimental setup in [1], where the authors considered QA only for the evaluation.
>
> ---
>
> **[W3&Q3] Lack of in-depth theoretical analysis, e.g., network design choice.**
>
> While our design choices were primarily guided by empirical analysis, we conducted an in-depth evaluation to determine the best architecture for our amortization and aggregation networks during development.
>
>
> For the amortization network, we explored three types of architectures: decoder-only, encoder-only, and encoder-decoder language models. Specifically, we considered (i) the GPT2 model and (ii) the T5 encoder with learnable tokens, where the input context is compressed into these tokens. The table below shows that the encoder-decoder model outperformed the other two alternatives, based on results from using GPT2-XL as the base LLM on the StreamingQA dataset. It is worth mentioning that our architecture follows the design carefully outlined in [4].
>
> \begin{array} {l cc}
> \hline
> \text{Amortization} & \text{EM} & \text{F1} \newline
> \hline
> \text{Encoder only (T5-encoder)} & 8.53 & 15.01 \newline
> \text{Decoder only (GPT2)} & 8.01 & 14.87 \newline
> \text{Encoder-Decoder (T5)} & \textbf{8.99} & \textbf{15.38} \newline
> \hline
> \end{array}
>
> For the aggregation network, we initially considered combining the amortization network (context compressed into PEFT modulation) with RAGs. However, we found that the aggregation approach provided better performance. Specifically, we trained an encoder-only model (T5-base encoder) to measure the similarity between PEFT modulations and the question for retrieving the modulation, subsequently adapting the base model. We report the results using GPT-XL as the base LLM on StreamingQA. Since the aggregation network attends to multiple documents and then predicts PEFT modulation, it is more likely to outperform RAG, which has a possibility of retrieving a wrong modulation.
>
> \begin{array} {l cc}
> \hline
> \text{Aggregation} & \text{EM} & \text{F1} \newline
> \hline
> \text{MAC (retrieve)} & 7.98 & 14.51 \newline
> \text{MAC (aggregate)} & \textbf{8.99} & \textbf{15.38} \newline
> \hline
> \end{array}
>
> ---
>
> **[Q1&Q4] How to handle storage overhead and redundancy of the memory bank.**
>
> First, we carefully remark that we have considered the storage overhead scenario in the main paper (in Figure 6, which is also reported in the table below for your convenience). Here, we consider a scenario with a memory bank size constraint by reducing the number of amortized contexts when it reaches 1,250 (where the total number of contexts is 1665). Here, we consider three simple yet effective schemes: i) random pruning, ii) randomly averaging two modulations, and iii) averaging two nearest neighbor (NN) modulations based on the cosine distance, where averaging nearest neighbor modulations shows quite effective preservation. This result indicates that redundant amortized context can be merged to improve storage efficiency while effectively maintaining the performance.
>
> \begin{array} {lcccc}\newline
> \hline
> & \text{Random pruning} & \text{Random averaging} & \text{NN averaging} & \text{Full memory} \newline
> \hline
> \text{F1} & 19.80 & 20.75 & 21.00 & 21.79 \newline
> \hline \newline
> \end{array}
>
> ---
>
> **Reference**\
> [1] Meta-Learning Online Adaptation of Language Models, EMNLP 2023\
> [2] Towards Continual Knowledge Learning of Language Models, ICLR 2022\
> [3] Meta-learning without memorization, ICLR 2020\
> [4] HyperTuning: Toward Adapting Large Language Models without Back-propagation, ICLR 2023

---

> > ### Comment · Area_Chair_voae · 2024-08-13
> >
> > Dear reviewer j5pf
> >
> > Could you please take a look at the responses of the authors and let us know your thoughts on them? Are you satisfied with the responses and do you have some updates on your comments?
> >
> > AC

---

### Official Review · Reviewer_kPwd · 2024-07-21

**Soundness:** 3
**Presentation:** 3
**Contribution:** 3
**Rating:** 6
**Confidence:** 3

**Summary:**

This paper presents a novel online adaptation framework (Memory of Amortised Contexts, MAC), which effectively solves the problem of rapid updating of large language models (LLMs). MAC successfully preserves the knowledge learned by the model during the original training phase and the new data streams through memory augmentation and efficient fine-tuning of parameters. Experimental results show that MAC outperforms existing online fine-tuning methods regarding online adaptive performance (Table 1 and Figure 3), time, and memory efficiency (Figures 2, 4, and 5). MAC can be combined with popular methods such as Retrieval-Augmented Generation (RAG) to improve performance further (Table 2).

**Strengths:**

1. In a time of rapidly updating information, MAC provides a practical approach to model updating that helps to keep language models current. With the proposed memory-efficient technique and forward propagation optimization, MAC significantly reduces the memory footprint and time consumption during online adaptation.
2. MAC effectively avoids the problem of catastrophic forgetting by constructing memory banks and ensures that the model integrates and utilizes old and new knowledge.
3. This paper validated the effectiveness of MAC through experiments with multiple datasets and different models, and the results are convincing.

**Weaknesses:**

1. Memory bank growth issues. As online adaptation proceeds, the size of the memory bank is likely to grow, which may pose a challenge for memory management. It is recommended that the authors explore more effective memory bank reduction techniques in future work.
2. The ability to generalize adaptations from different domains remains unknown. Often, online learning may not be the same type of task (e.g., knowledge answering and coding), and it remains unclear whether the current method can cope with such scenarios.

**Questions:**

1. The current experiments are mainly validation on small models, but according to the author's description llama was also trained for 50 epochs, which seems to be problematic?
2. The current fine-tuning is based on P-Tuning. Has there been any consideration of comparing more parameter-efficient fine-tuning methods, such as Lora fine-tuning?
3. Is it possible to add scenario experiments for online learning related to different tasks?

**Limitations:**

Yes

---

> ### Author Rebuttal · Authors · 2024-08-07
>
> Dear reviewer kPwd,
>
> We sincerely appreciate your efforts and insightful comments to improve the manuscript.\
> We respond to each of your comments one-by-one in what follows.
>
> ---
>
> **[W1] Memory bank growth issues.**
>
> It is true that one possible limitation of MAC can be the growing size of the memory bank during adaptation.
>
> However, we would like to remark that we have already considered such a scenario in the main paper by constraining the memory bank size (in Figure 6 and also in the table below). Specifically, we reduce the number of amortized contexts when it reaches the memory constraint of 1,250 (where the total number of contexts is 1665). Here, we consider three simple yet effective schemes: i) random pruning, ii) randomly averaging two modulations, and iii) averaging two nearest neighbor (NN) modulations based on the cosine distance, where averaging nearest neighbor modulations shows quite effective preservation.
>
> \begin{array} {lcccc}\newline \hline & \text{Random pruning} & \text{Random averaging} & \text{NN averaging} & \text{Full memory} \newline \hline \text{F1} & 19.80 & 20.75 & 21.00 & 21.79 \newline \hline\newline \end{array}
>
> ---
>
> **[W2] Ability to adapt to different tasks (e.g., coding) and domains.**
>
> First, we clarify that the major reason we mainly focused on the QA task is that it is a conventional evaluation protocol for online learning LMs [1,2,3], as evaluating the updated (or preserved) knowledge is non-trivial for other tasks. In this regard, we followed the same experimental setup in [1], where the authors considered QA only for the evaluation.
>
> Nevertheless, we would like to emphasize the strong adaptation ability of MAC to other domains in QA (in Table 4/also in the table below), thus showing the possibility of adapting to different tasks when the training corpus increases. Here, we show that MAC trained on the StreamingQA dataset can be used for online adaptation of different QA datasets. As shown in the table below, MAC outperforms CaMeLS in F1 score. It is worth noting that the meta-learning performance scales as the training distribution is more diverse [4], hence, we believe training MAC on larger datasets and tasks will further improve the generalization.
>
> \begin{array} {l cc} \hline \text{StreamQA}\to & \text{SQuAD} & \text{ArchivalQA} \newline \hline \text{CaMeLS} & 8.63 & 13.43 \newline \text{MAC} & \textbf{10.47} & \textbf{13.73} \newline \hline \end{array}
>
> ---
>
> **[Q1] Mainly validated on small models.**
>
> First, we want to clarify that we followed the setup from the prior paper [1], where we actually conducted experiments on a larger model compared to the previous work (i.e., Llama2 7B). This was possible because MAC is more efficient in terms of both time and memory compared to the baseline [1]. For instance, the baseline could not train on Llama2 7B within the given memory constraint of 80GB with a batch size of 1, even with 4-bit quantization.
>
> ---
>
> **[Q2] Other types of PEFT methods.**
>
> Thank you for bringing this up. During our initial development, we also considered LoRA as an alternative. However, we found that P-tuning v2 outperformed LoRA when training GPT2-XL on the StreamingQA dataset. This aligns with findings from previous work [5], which also observed that P-tuning outperforms LoRA when using amortization. Additionally, P-tuning allows for efficient batch computation, enabling a single forward pass of the LLM with different modulations. In contrast, LoRA requires separate forward passes for each modulation, which increases the training time. For these reasons, we chose to use P-tuning throughout our paper.
>
> \begin{array} {l cc}
> \hline
> \text{PEFT type} & \text{EM} & \text{F1} \newline
> \hline
> \text{LoRA} & 8.67 & 15.15 \newline
> \text{P-tuning v2} & \textbf{8.99} & \textbf{15.38} \newline
> \hline
> \end{array}
>
> ---
>
> **[Q3] Other scenarios than QA.**
>
> We carefully remark that we already considered an additional scenario (i.e., the language modeling) rather than QA in Table 5, where we outperformed other online finetuning baselines. For your convenience, we have presented the table below. Specifically, we adapt the LLM on a stream of documents, then give the initial 10% of the document as input and measure the perplexity of the remaining documents (the initial document is equivalent to a question in the QA task). Here, we measure the perplexity of the predicted text on two document sets: i) the documents used for LLM adaptation to measure knowledge preservation and ii) unseen documents to measure the generalization, where MAC has outperformed the baselines in both cases.
>
> \begin{array} {lcc}\newline \hline &\text{Adapted documents} &\text{Unseen documents} \newline \hline \text{Uniform} & 11.43 & 13.89 \newline \text{SSM} & 27.87 & 29.69 \newline \text{CaMeLS} & 11.31 & 14.77 \newline \text{MAC (Ours)} & \textbf{10.91} & \textbf{12.71} \newline \hline\newline \end{array}
>
> ---
>
> **Reference**\
> [1] Meta-Learning Online Adaptation of Language Models, EMNLP 2023\
> [2] Towards Continual Knowledge Learning of Language Models, ICLR 2022\
> [3] TemporalWiki: A Lifelong Benchmark for Training and Evaluating Ever-Evolving Language Models, EMNLP 2022\
> [4] Meta-learning without memorization, ICLR 2020\
> [5] HyperTuning: Toward Adapting Large Language Models without Back-propagation, ICLR 2023

---

> > ### Comment · Reviewer_kPwd · 2024-08-13
> >
> > Thank you for your response. I keep my score unchanged.

---

> > > ### Author Response · Authors · 2024-08-14
> > > **Thank you very much for the response**
> > >
> > > Dear Reviewer kPwd
> > >
> > > Thank you for letting us know! We are delighted to hear that our rebuttal addressed your questions well.\
> > > If you have any further questions or suggestions, please do not hesitate to let us know.
> > >
> > > Thank you very much,\
> > > Authors

---

### Author Rebuttal · Authors · 2024-08-07

Dear reviewers and AC,

We sincerely appreciate your valuable time and effort spent reviewing our manuscript.

As reviewers highlighted, we believe our paper provides a novel (kPwd,j5pf), efficitent (all reviewers) yet effective (kPwd, pwuf) framework for online adaptation of LLMs followed by a clear presentation (all reviewers).

We appreciate your constructive comments on our manuscript. In the attached pdf, we have run the following additional experiment to clarify the reviewer's comment:
- Memory efficiency and performance curve of RAG and MAC (combined with RAG), in Figure 1
- Comparison between RAG and MAC (combined with RAG), in Table 1

We strongly believe that MAC can be a useful addition to the NeurIPS community, in particular, due to the enhanced manuscript by reviewers’ comments helping us better deliver the effectiveness of our method.

Thank you very much!\
Authors

---

### Decision · Program_Chairs · 2024-09-25

**Decision:**

Accept (poster)

**Comment:**

This paper introduces a novel framework for efficient online adaptation of LLMs. The method uses feature extraction and memory augmentation to compress and store information from new documents in a memory bank, which can be retrieved when answering questions. The method employs amortization-based meta-learning, allowing for efficient modulation learning through a single forward pass without requiring gradient updates during testing.

The reviewers generally agree that the proposed method presents a novel and efficient approach to online adaptation of LLMs. The method's ability to outperform baselines while maintaining memory efficiency is particularly noteworthy. The authors have addressed many of the reviewers' concerns in their rebuttal, such as clarifying the fairness of comparisons, providing additional experimental results, and explaining the impact of quantization on larger models.